# DP-SGD Without Clipping:
# The Lipschitz Neural Network Way

## Abstract

State-of-the-art approaches for training Differentially Private (DP) Deep Neural Networks (DNN) faces difficulties to estimate tight bounds on the sensitivity of the network's layers, and instead rely on a process of per-sample gradient clipping. This clipping process not only biases the direction of gradients but also proves costly both in memory consumption and in computation. To provide sensitivity bounds and bypass the drawbacks of the clipping process, our theoretical analysis of Lipschitz constrained networks reveals an unexplored link between the Lipschitz constant with respect to their input and the one with respect to their parameters. By bounding the Lipschitz constant of each layer with respect to its parameters we guarantee DP training of these networks. This analysis not only allows the computation of the aforementioned sensitivities at scale but also provides leads on to how maximize the gradient-to-noise ratio for fixed privacy guarantees. To facilitate the application of Lipschitz networks and foster robust and certifiable learning under privacy guarantees, we provide a Python package that implements building blocks allowing the construction and private training of such networks.

## 1 Introduction

Machine learning relies more than ever on foundational models, and such practices raise questions about privacy. Differential privacy allows to develop methods for training models that preserve the privacy of individual data points in the training set. The field seeks to enable deep learning on sensitive data, while ensuring that models do not inadvertently memorize or reveal specific details about individual samples in their weights. This involves incorporating privacy-preserving mechanisms into the design of deep learning architectures and training algorithms, whose most popular example is Differentially Private Stochastic Gradient Descent (DP-SGD) [1]. One main drawback of classical DP-SGD methods is that they require costly per-sample backward processing and gradient clipping. In this paper, we offer a new method that unlocks fast differentially private training through the use of Lipschitz constrained neural networks. Additionally, this method offers new opportunities for practitioners that wish to easily "DP-fy" [2] the training procedure of a deep neural network.

**Differential privacy fundamentals.** Informally, differential privacy is a *definition* that quantifies how much the change of a single sample in a dataset affects the range of a stochastic function (here the DP training), called *mechanism* in this context. This quantity can be bounded in an inequality involving two parameters $\epsilon$ and $\delta$. A mechanism fulfilling such inequality is said $(\epsilon, \delta)$-DP (see Definition 1). This definition is universally accepted as a strong guarantee against privacy leakages under various scenarii, including data aggregation or post-processing [3]. A popular rule of thumb suggests using $\epsilon \leq 10$ and $\delta < \frac{1}{N}$ with $N$ the number of records [2] for mild guarantees. In practice, most classic algorithmic procedures (called *queries* in this context) do not readily fulfill the definition for useful values of $(\epsilon, \delta)$, in particular the deterministic ones: randomization is mandatory. This randomization

```
model = DP_Sequential( # step 1: use DP_Sequential to build a model
    [
        # step 2: add Lipschitz layers of known sensitivity
        DP_BoundedInput(input_shape=(28, 28, 1), upper_bound=20.),
        DP_SpectralConv2D(filters=16, kernel_size=3, use_bias=False),
        DP_GroupSort(2),
        DP_Flatten(),
        DP_SpectralDense(10),
    ],
    noise_multiplier = 1.2, # step 3: choose DP parameters
    sampling_probability = batch_size / dataset_size,
) # step 4: compile the model, and choose any first order optimizer
model.compile(loss=DP_Crossentropy(), optimizer=Adam(1e-3))
model.fit( # step 5: train the model and measure the DP guarantees
    train_dataset, validation_data=val_dataset,
    epochs=num_epochs, callbacks=[DP_Accountant()]
)
```

Figure 1: **An example of usage of our framework**, illustrating how to create a small Lipschitz VGG and how to train it under $(\epsilon, \delta)$-DP guarantees while reporting $(\epsilon, \delta)$ values.

comes at the expense of "utility", i.e the usefulness of the output for downstream tasks [4]. The goal is then to strike a balance between privacy and utility, ensuring that the released information remains useful and informative for the intended purpose while minimizing the risk of privacy breaches. The privacy/utility trade-off yields a Pareto front, materialized by plotting $\epsilon$ against a measurement of utility, such as validation accuracy for a classification task.

**Private gradient descent.** The SGD algorithm consists of a sequence of queries that (i) take the dataset in input, sample a minibatch from it, and return the gradient of the loss evaluated on the minibatch, before (ii) performing a descent step following the gradient direction. The sensitivity (see Definition 2) of SGD queries is proportional to the norm of the per-sample gradients. DP-SGD turns each query into a Gaussian mechanism by perturbing the gradients with a noise $\zeta$. The upper bound on gradient norms is generally unknown in advance, which leads practitioners to clip it to $C > 0$, in order to bound the sensitivity manually. This is problematic for several reasons: **1.** Hyper-parameter search on the broad-range clipping value $C$ is required to train models with good privacy/utility trade-offs [5], **2.** The computation of per-sample gradients is expensive: DP-SGD is usually slower and consumes more memory than vanilla SGD, in particular for the large batch sizes often used in private training [6], **3.** Clipping the per-sample gradients biases their average [7]. This is problematic as the average direction is mainly driven by misclassified examples, that carry the most useful information for future progress.

**An unexplored approach: Lipschitz constrained networks.** We propose to train neural networks for which the parameter-wise gradients are provably and analytically bounded during the whole training procedure, in order to get rid of the clipping process. This allows for rapid training of models without a need for tedious hyper-parameter optimization.

The main reason why this approach has not been experimented much in the past is that upper bounding the gradient of neural networks is often intractable. However, by leveraging the literature of Lipschitz constrained networks [8], we show that these networks allows to estimate their gradient bound. This yields tight bounds on the sensitivity of SGD steps, making their transformation into Gaussian mechanisms inexpensive - hence the name **Clipless DP-SGD**.

Informally, the Lipschitz constant quantifies the rate at which the function's output varies with respect to changes in its input. A Lipschitz constrained network is one in which its weights and activations are constrained such that it can only represent $l$-Lipschitz functions. In this work, we will focus our attention on feed-forward networks (refer to Definition 3). Note that the most common architectures, such as Convolutional Neural Networks (CNNs), Fully Connected Networks (FCNs), Residual Networks (ResNets), or patch-based classifiers (like MLP-Mixers), all fall under the category of feed-forward networks. We will also tackle the particular case of Gradient Norm Preserving (GNP) networks, a subset of Lipschitz networks that enjoy tighter bounds (see appendix).

**Contributions**

While the properties of Lipschitz constrained networks regarding their inputs are well explored, the properties with respect to its parameters remain non-trivial. This work provides a first step to fill this gap: our analysis shows that under appropriate architectural constraints, a $l$-Lipschitz network has a tractable, finite Lipschitz constant with respect to its parameters. We prove that this Lipschitz constant allows for easy estimation of the sensitivity of the gradient computation queries. The prerequisite and details of the method to compute the sensitivities are explained in Section 2.

Our contributions are the following:

1. We extend the field of applications of Lipschitz constrained neural networks. So far the literature focused on Lipschitzness with respect to the *inputs*: we extend the framework to **compute the Lipschitzness with respect to the parameters**. This is exposed in Section 2.

2. We propose a **general framework to handle layer gradient steps as Gaussian mechanisms** that depends on the loss and the model structure. Our framework covers widely used architectures, including VGG and ResNets.

3. We show that SGD training of deep neural networks can be achieved **without gradient clipping** using Lipschitz layers. This allows the use of larger networks and larger batch sizes, as illustrated by our experiments in Section 4.

4. We establish connections between **Gradient Norm Preserving** (GNP) networks and **improved privacy/utility trade-offs** (Section 3.1).

5. Finally, a **Python package**[1] companions the project, with pre-computed Lipschitz constant and noise for each layer type, ready to be forked on any problem of interest (Section 3.2).

## 1.1 Differential Privacy and Lipschitz Networks

The definition of DP relies on the notion of neighboring datasets, i.e datasets that vary by at most one example. We highlight below the central tools related to the field, inspired from [9].

**Definition 1** (($\epsilon, \delta$)-Differential Privacy). *A labeled dataset $\mathcal{D}$ is a finite collection of input/label pairs $\mathcal{D} = \{(x_1, y_1), (x_2, y_2), \ldots..(x_N, y_N)\}$. Two datasets $\mathcal{D}$ and $\mathcal{D}'$ are said to be neighboring for the "replace-one" relation if they differ by at most one sample: $\mathcal{D}' = \mathcal{D} \cup \{(x_i', y_i')\} \setminus \{(x_i, y_i)\}$. Let $\epsilon$ and $\delta$ be two non-negative scalars. A mechanism $\mathcal{A}$ is ($\epsilon, \delta$)-DP if for any two neighboring datasets $\mathcal{D}$ and $\mathcal{D}'$, and for any $S \subseteq range(\mathcal{A})$:*

$$\mathbb{P}[\mathcal{A}(\mathcal{D}) \in S] \leq e^{\epsilon} \times \mathbb{P}[\mathcal{A}(\mathcal{D}') \in S] + \delta. \tag{1}$$

A cookbook to create a ($\epsilon, \delta$)-DP mechanism from a query is to compute its *sensitivity* $\Delta$ (see Definition 2), and to perturb its output by adding a Gaussian noise of predefined variance $\zeta^2 = \Delta^2 \sigma^2$, where the ($\epsilon, \delta$)-DP guarantees depends on $\sigma$. This yields what is called a *Gaussian mechanism* [3].

**Definition 2** ($l_2$-sensitivity). *Let $\mathcal{M}$ be a query mapping from the space of the datasets to $\mathbb{R}^p$. Let $\mathcal{N}$ be the set of all possible pairs of neighboring datasets $\mathcal{D}, \mathcal{D}'$. The $l_2$ sensitivity of $\mathcal{M}$ is defined by:*

$$\Delta(\mathcal{M}) = \max_{\mathcal{D}, \mathcal{D}' \in \mathcal{N}} \|\mathcal{M}(D) - \mathcal{M}(D')\|_2. \tag{2}$$

**Differentially Private SGD.** The classical algorithm keeps track of ($\epsilon, \delta$)-DP values with a *moments accountant* [1] which allows to keep track of privacy guarantees at each epoch, by composing different sub-mechanisms. For a dataset with $N$ records and a batch size $b$, it relies on two parameters: the sampling ratio $p = \frac{b}{N}$ and the "noise multiplier" $\sigma$ defined as the ratio between effective noise strength $\zeta$ and sensitivity $\Delta$. Bounds on gradient norm can be turned into bounds on sensitivity of SGD queries. In "replace-one" policy for ($\epsilon, \delta$)-DP accounting, if the gradients are bounded by $K > 0$, the sensitivity of the gradients averaged on a minibatch of size $b$ is $\Delta = 2K/b$..

Crucially, the algorithm requires a bound on $\|\nabla_\theta \mathcal{L}(\hat{y}, y)\|_2 \leq K$. The whole difficulty lies in bounding tightly this value in advance for neural networks. Currently, gradient clipping serves as a patch to circumvent the issue [1]. Unfortunately, clipping individual gradients in the batch is costly and will bias the direction of their average, which may induce underfitting [7].

---

[1]Code and documentation are given as supplementary material during review process.

117 **Lipschitz constrained networks.** Our proposed solution comes from the observation that the norm
118 of the gradient and the Lipschitz constant are two sides of the same coin. The function $f : \mathbb{R}^m \to \mathbb{R}^n$
119 is said $l$-Lipschitz for $l_2$ norm if for every $x, y \in \mathbb{R}^m$ we have $\|f(x) - f(y)\|_2 \leq l\|x - y\|_2$. Per
120 Rademacher's theorem [10], its gradient is bounded: $\|\nabla_x f\| \leq l$. Reciprocally, continuous functions
121 gradient bounded by $l$ are $l$-Lipschitz.

122 In Lipschitz networks, the literature has predominantly concentrated on investigating the control
123 of Lipschitzness with respect to the inputs (i.e bounding $\nabla_x f$), primarily motivated by concerns
124 of robustness [11]. However, in this work, we will demonstrate that it is also possible to control
125 Lipschitzness with respect to parameters (i.e bounding $\nabla_\theta f$), which is essential for ensuring privacy.
126 Our first contribution will point out the tight link that exists between those two quantities.

127 **Definition 3** (Lipschitz feed-forward neural network). *A feedforward neural network of depth $D$,*
128 *with input space $\mathcal{X} \subset \mathbb{R}^n$, output space $\mathcal{Y} \subset \mathbb{R}^K$ (e.g logits), and parameter space $\Theta \subset \mathbb{R}^p$, is a*
129 *parameterized function $f : \Theta \times \mathcal{X} \to \mathcal{Y}$ defined by the sequential composition of layers $f_d$:*

$$f(\theta, x) := (f_D(\theta_d) \circ \ldots \circ f_2(\theta_2) \circ f_1(\theta_1))(x). \tag{3}$$

130 *The parameters of the layers are denoted by $\theta = (\theta_d)_{1 \leq d \leq D} \in \Theta$. For affine layers, it corresponds*
131 *to bias and weight matrix $\theta_d = (W_d, b_d)$. For activation functions, there is no parameters: $\theta_d = \varnothing$.*

132 *Lipschitz networks are feed-forward networks, with the additionnal constraint that each*
133 *layer $x_d \mapsto f_d(\theta_d, x_d) := y_d$ is $l_d$-Lipschitz for all $\theta_d$. Consequently, the function $x \mapsto f(\theta, x)$*
134 *is $l$-Lipschitz with $l = l_1 \times \ldots \times l_d$ for all $\theta \in \Theta$.*

135 In practice, this is enforced by using activations with Lipschitz constant $l_d$, and by applying a con-
136 straint $\Pi : \mathbb{R}^p \to \Theta$ on the weights of affine layers. This corresponds to spectrally normalized matri-
137 ces [12, 13], since for affine layers we have $l_d = \|W_d\|_2 := \max_{\|x\| \leq 1} \|W_d x\|_2$ hence $\Theta = \{\|W_d\| \leq l_q\}$.

138 The seminal work of [8] proved that universal approximation in the set of $l$-Lipschitz functions was
139 achievable by this family of architectures. Concurrent approaches are based on regularization (like
140 in [14, 15, 16]) but they fail to produce formal guarantees. While they have primarily been studied in
141 the context of adversarial robustness [11, 17], recent works have revealed additional properties of
142 these networks, such as improved generalization [13, 18]. However, the properties of their parameter
143 gradient $\nabla_\theta f(\theta, x)$ remain largely unexplored.

## 2 Clipless DP-SGD with $l$-Lipschitz networks

145 Our framework consists of **1.** a method that computes the maximum gradient norm of a network with
146 respect to its parameters to obtain a *per-layer* sensitivity $\Delta_d$, **2.** a moments accountant that relies on
147 the per-layer sensitivities to compute $(\epsilon, \delta)$-DP guarantees. The method 1. is based on the recursive
148 formulation of the chain rule involved in backpropagation, while 2. keeps track of $(\epsilon, \delta)$-DP values
149 with RDP accounting. It requires some natural assumptions that we highlight below.

150 **Requirement 1** (Lipschitz loss.). *The loss function $\hat{y} \mapsto \mathcal{L}(\hat{y}, y)$ must be $L$-Lipschitz with respect to*
151 *the logits $\hat{y}$ for all ground truths $y \in \mathcal{Y}$. This is notably the case of Categorical Softmax-Crossentropy.*

152 The Lipschitz constants of common classification losses can be found in the appendix.

153 **Requirement 2** (Bounded input). *There exists $X_0 > 0$ such that for all $x \in \mathcal{X}$ we have $\|x\| \leq X_0$.*

154 While there exist numerous approaches for the parametrization of Lipschitz networks (e.g differen-
155 tiable re-parametrization [19, 8], optimization over matrix manifolds [20] or projections [21]), our
156 framework only provides sensitivity bounds for projection-based algorithms (see appendix).

157 **Requirement 3** (Lipschitz projection). *The Lipschitz constraints must be enforced with a projection*
158 *operator $\Pi : \mathbb{R}^p \to \Theta$. This corresponds to Tensorflow [22] `constraints` and Pytorch [23] `hooks`.*
159 *Projection is a post-processing of private gradients: it induces no privacy leakage [3].*

160 To compute the per-layer sensitivities, our framework mimics the backpropagation algorithm, where
161 *Vector-Jacobian* products (VJP) are replaced by *Scalar-Scalar* products of element-wise bounds. For
162 an arbitrary layer $x_d \mapsto f_d(\theta_d, x_d) := y_d$ the operation is sketched below:

$$\underbrace{\nabla_{x_d}\mathcal{L} := (\nabla_{y_d}\mathcal{L})\frac{\partial f_d}{\partial x_d}}_{\text{Vector-Jacobian product: backpropagate gradients}} \implies \underbrace{\|\nabla_{x_d}\mathcal{L}\|_2 \leq \|\nabla_{y_d}\mathcal{L}\|_2 \times \left\|\frac{\partial f_d}{\partial x_d}\right\|_2}_{\text{Scalar-Scalar product: backpropagate bounds}}. \tag{4}$$

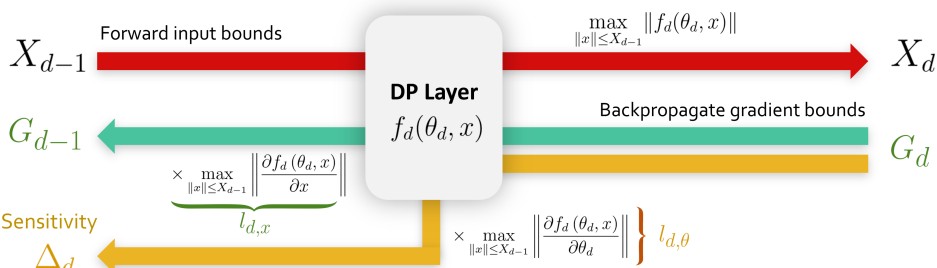

Figure 2: **Backpropagation for bounds**, Algorithm 1. Compute the per-layer sensitivity $\Delta_d$.

The notation $\|\cdot\|_2$ must be understood as the spectral norm for Jacobian matrices, and the Euclidean norm for gradient vectors. The scalar-scalar product is inexpensive. For Lipschitz layers the spectral norm of the Jacobian $\|\frac{\partial f}{\partial x}\|$ is kept constant during training with projection operator $\Pi$. The bound of the gradient with respect to the parameters then takes a simple form:

$$\|\nabla_{\theta_d}\mathcal{L}\|_2 = \|\nabla_{y_d}\mathcal{L}\|_2 \times \left\|\frac{\partial f_d}{\partial \theta_d}\right\|_2. \tag{5}$$

Once again the operation is inexpensive. The upper bound $\left\|\frac{\partial f}{\partial \theta}\right\|_2$ typically depends on the supremum of $\|x_d\|_2$, that can also be analytically bounded, as exposed in the following section.

## 2.1 Backpropagation for bounds

The pseudo-code of **Clipless DP-SGD** is sketched in Algorithm 2. The algorithm avoids clipping by computing a *per-layer* bound on the element-wise gradient norm. The computation of this *per-layer* bound is described by Algorithm 1 (graphically explained in Figure 2). Crucially, it requires to compute the spectral norm of the Jacobian of each layer with respect to input and parameters.

**Input bound propagation (line 2).** We compute $X_d = \max_{\|x\|\le X_{d-1}} \|f_d(x)\|_2$. For activation functions it depends on their range. For linear layers, it depends on the spectral norm of the operator itself. This quantity can be computed with SVD or Power Iteration [24, 19], and constrained during training using projection operator $\Pi$. In particular, it covers the case of convolutions, for which tight bounds are known [25]. For affine layers, it additionally depends on the amplitude of the bias $\|b_d\|$.

**Remark 1** (Tighter bounds in literature.)*. Although libraries such as Decomon [26] or auto-LiRPA [27] provide tighter bounds $X_d$ via linear relaxations [28, 29], our approach is capable of delivering practically tighter bounds than worst-case scenarios thanks to the projection operator $\Pi$, while also being significantly less computationally expensive. Moreover, hybridizing our method with scalable certification methods can be a path for future extensions.*

**Computing maximum gradient norm (line 6).** We bound the Jacobian $\frac{\partial f_d(\theta_d,x)}{\partial \theta_d}$. In neural networks, the parameterized layers $f(\theta,x)$ (fully connected, convolutions) are bilinear operators. Hence we typically obtain bounds of the form:

$$\left\|\frac{\partial f_d(\theta_d,x)}{\partial \theta_d}\right\|_2 \le K(f_d,\theta_d)\|x\|_2 \le K(f_d,\theta_d)X_{d-1}, \tag{6}$$

where $K(f_d,\Theta_d)$ is a constant that depends on the nature of the operator. $X_{d-1}$ is obtained in line 2 with input bound propagation. Values of $K(f_d,\theta_d)$ for popular layers are pre-computed in the library.

**Backpropagate cotangeant vector bounds (line 7).** We bound the Jacobian $\frac{\partial f_d(\theta_d,x)}{\partial x}$. For activation functions this value can be hard-coded, while for affine layers it is the spectral norm of the linear operator. Like before, this value is constrained with projection operator $\Pi$.

## 2.2 Privacy accounting for Clipless DP-SGD

Two strategies are available to keep track of $(\epsilon, \delta)$ values as the training progresses, based on accounting either a per-layer "local" sensitivity, either by aggregating them into a "global" sensitivity.

---

**Algorithm 1 Backpropagation for Bounds**$(f, X)$

---

**Input**: Feed-forward architecture $f(\theta, \cdot) = f_D(\theta_D, \cdot) \circ \ldots \circ f_1(\theta_1, \cdot)$
**Input**: Weights $\theta = (\theta_1, \theta_2, \ldots \theta_D)$, input bound $X_0$

  1: **for all** layers $1 \leq d \leq D$ **do**
  2:     $X_d \leftarrow \max_{\|x\| \leq X_{d-1}} \|f_d(\theta_d, x)\|_2.$             ▷ Input bounds propagation
  3: **end for**
  4: $G \leftarrow L/b.$             ▷ Lipschitz constant of the loss for batchsize b
  5: **for all** layers $D \geq d \geq 1$ **do**
  6:     $\Delta_d \leftarrow G \max_{\|x\| \leq X_{d-1}} \|\frac{\partial f_d(\theta_d, x)}{\partial \theta_d}\|_2.$     ▷ Compute sensitivity from gradient norm
  7:     $G \leftarrow G \max_{\|x\| \leq X_{d-1}} \|\frac{\partial f_d(\theta_d, x)}{\partial x}\|_2 = G l_d.$     ▷ Backpropagate cotangeant vector bounds
  8: **end for**
  9: **return** sensitivities $\Delta_1, \Delta_2 \ldots, \Delta_D$

---

**Algorithm 2 Clipless DP-SGD** with **local** sensitivity accounting

---

**Input**: Feed-forward architecture $f(\theta, \cdot) = f_D(\theta_D, \cdot) \circ \ldots \circ f_1(\theta_1, \cdot)$
**Input**: Initial weights $\theta = (\theta_1, \theta_1, \ldots \theta_D)$, learning rate $\eta$, noise multiplier $\sigma$.

  1: **repeat**
  2:     $\Delta_1, \Delta_2 \ldots \Delta_D \leftarrow$ **Backpropagation for Bounds**$(f, X)$.
  3:     Update Moment Accountant state with **local** sensitivities $\Delta_1, \Delta_2, \ldots \Delta_d$.
  4:     Sample a batch $\mathcal{B} = \{(x_1, y_1), (x_2, y_2), \ldots, (x_b, y_b)\}$.
  5:     Compute per-layer averaged gradient: $g_d := \frac{1}{b} \sum_{i=1}^{b} \nabla_{\theta_d} \mathcal{L}(f(\theta, x_i), y_i)).$
  6:     Sample local noise: $\zeta_d \sim \mathcal{N}(0, \sigma \Delta_d)$.
  7:     Perform noisified gradient step: $\theta_d \leftarrow \theta_d - \eta(g_d + \zeta_d)$.
  8:     Enforce Lipschitz constraint with projection: $\theta_d \leftarrow \Pi(\theta_d)$.
  9: **until** privacy budget $(\epsilon, \delta)$-DP budget has been reached.

---

**The "global" strategy.** Illustrated in the appendix, this strategy simply aggregates the individual sensitivities $\Delta_d$ of each layer to obtain the global sensitivity of the whole gradient vector $\Delta = \sqrt{\sum_d \Delta_d^2}$. The origin of the clipping-based version of this strategy can be traced back to [30]. With noise variance $\sigma^2 \Delta^2$ we recover the accountant that comes with DP-SGD. It tends to overestimate the true sensitivity (in particular for deep networks), but its implementation is straightforward with existing tools.

**The "local" strategy.** Recall that we are able to characterize the sensitivity $\Delta_d$ of every layer of the network. Hence, we can apply a different noise to each of the gradients. We dissect the whole training procedure in Figure 3. At same noise multiplier $\sigma$, it tends to produce a higher value of $\epsilon$ per epoch than "global" strategy, but has the advantage over the latter to add smaller effective noise $\zeta$ to each weight.

We rely on the `autodp`[2] library [32, 33, 34] as it uses the Renyi Differential Privacy (RDP) adaptive composition theorem [35, 36], that ensures tighter bounds than naive DP composition.

## 3   From theory to practice

Beyond the application of Algorithms 1 and 2, our framework provides numerous opportunities to enhance our understanding of prevalent techniques identified in the literature. An in-depth exploration of these is beyond the scope of this work, so we focus on giving insights on promising tracks based on our theoretical analysis. In particular, we discuss how the tightness of the bound provided by Algorithm 1 can be influenced by working on the architecture, the input pre-processing and the loss post-processing.

---

[2]`https://github.com/yuxiangw/autodp` distributed under Apache License 2.0 licence.

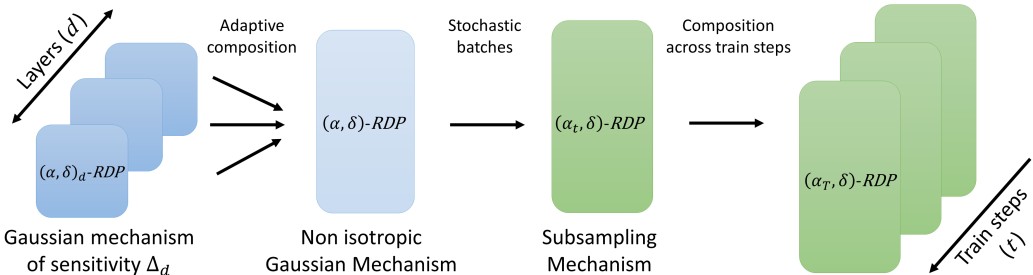

Figure 3: **Accountant for locally enforced differential privacy. (i)** The gradient query for each layer is turned into a Gaussian mechanism [9], **(ii)** their composition at the scale of the whole network is a non isotropic Gaussian mechanism, **(iii)** that benefits from amplification via sub-sampling [31], **(iv)** the train steps are composed over the course of training.

### 3.1 Gradient Norm Preserving networks

We can manually derive the bounds obtained from Algorithm 2 across diverse configurations. Below, we conduct a sensitivity analysis on $l$-Lipschitz networks.

**Theorem (informal) 1. Gradient Norm of Lipschitz Networks.** *Assume that every layer $f_d$ is $K$-Lipschitz, i.e $l_1 = \cdots = l_D = K$. Assume that every bias is bounded by $B$. We further assume that each activation is centered in zero (e.g ReLU, tanh, GroupSort). We recall that $\theta = [\theta_1, \theta_2, \ldots \theta_D]$. Then the global upper bound of Algorithm 2 can be expanded analytically.*

***1. If $K < 1$ we have:*** $\|\nabla_\theta \mathcal{L}(f(\theta, x), y)\|_2 = \mathcal{O}\left(L\left(K^D(X_0 + B) + 1\right)\right).$

*Due to the $K^D \ll 1$ term this corresponds to a vanishing gradient phenomenon [37]. The output of the network is essentially independent of its input, and the training is nearly impossible.*

***2. If $K > 1$ we have:*** $\|\nabla_\theta \mathcal{L}(f(\theta, x), y)\|_2 = \mathcal{O}\left(LK^D(X_0 + B)\right).$

*Due to the $K^D \gg 1$ term this corresponds to an exploding gradient phenomenon [38]. The upper bound becomes vacuous for deep networks: the added noise $\zeta$ is at risk of being too high.*

***3. If $K = 1$ we have:*** $\|\nabla_\theta \mathcal{L}(f(\theta, x), y)\|_2 = \mathcal{O}\left(L\left(X_0 + \sqrt{D} + \sqrt{BX_0}D + BD^{3/2}\right)\right),$

*which for linear layers without biases further simplify to $\mathcal{O}(L(X_0 + \sqrt{D}))$.*

The formal statement can be found in appendix. From Theorem 1 we see that most favorable bounds are achieved by 1-Lipschitz neural networks with 1-Lipschitz layers. In classification tasks, they are not less expressive than conventional networks [18]. Hence, this choice of architecture is not at the expense of utility. Moreover an accuracy/robustness trade-off exists, determined by the choice of loss function [18]. However, setting $K = 1$ merely ensures that $\|\nabla_x f\| \leq 1$, and in the worst-case scenario we have $\|\nabla_x f\| < 1$ almost everywhere. This could result in a situation where the bound of case 3 in Theorem 1 is not tight, leading to an underfitting regime as in case $K < 1$. With Gradient Norm Preserving (GNP) networks [17], we expect to mitigate this issue.

**Controlling $K$ with Gradient Norm Preserving (GNP) networks.** GNP networks are 1-Lipschitz neural networks with the additional constraint that the Jacobian of layers consists of orthogonal matrices. They fulfill the Eikonal equation $\left\|\frac{\partial f_d(\theta_d, x_d)}{\partial x_d}\right\|_2 = 1$ for any intermediate activation $f_d(\theta_d, x_d)$. Without biases these networks are also norm preserving: $\|f(\theta, x)\| = \|x\|$.

As a consequence, the gradient of the loss with respect to the parameters is easily bounded by

$$\|\nabla_{\theta_d} \mathcal{L}\| = \|\nabla_{y_D} \mathcal{L}\| \times \left\|\frac{\partial f_d(\theta_d, x_d)}{\partial \theta_d}\right\|, \tag{7}$$

which for weight matrices $W_d$ further simplifies to $\|\nabla_{W_d} \mathcal{L}\| \leq \|\nabla_{y_D} \mathcal{L}\| \times \|f_{d-1}(\theta_{d-1}, x_{d-1})\|$. We see that this upper bound crucially depends on two terms than can be analyzed separately. On one hand, $\|f_{d-1}(\theta_{d-1}, x_{d-1})\|$ depends on the scale of the input. On the other, $\|\nabla_{y_D} \mathcal{L}\|$ depends on the loss, the predictions and the training stage. We show below how to intervene on these two quantities.

**Remark 2** (Implementation of GNP Networks). *In practice, GNP are parametrized with GroupSort activation [8, 39], Householder activation [40], and orthogonal weight matrices [17, 41]. Strict orthogonality is challenging to enforce, especially for convolutions for which it is still an active research area (see [42, 43, 44, 45, 46] and references therein). Our line of work traces an additional motivation for the development of GNP and the bounds will strengthen as the field progresses.*

**Controlling $X_0$ with input pre-processing.** The weight gradient norm $\|\nabla_{\theta_d}\mathcal{L}\|$ indirectly depends on the norm of the inputs. This observation implies that the pre-processing of input data significantly influences the bounding of sensitivity. Multiple strategies are available to keep the input's norm under control: projection onto the ball ("norm clipping"), or projection onto the sphere ("normalization"). In the domain of natural images for instance, this result sheds light on the importance of color space such as RGB, HSV, YIQ, YUV or Grayscale. These strategies are natively handled by our library.

**Controlling $L$ with the hybrid approach, loss gradient clipping.** As training progresses, the magnitude of $\|\nabla_f\mathcal{L}\|$ tends to diminish when approaching a local minima, quickly falling below the upper bound and diminishing the gradient norm to noise ratio. To circumvent the issue, the gradient clipping strategy is still available in our framework. Crucially, instead of clipping the parameter gradient $\nabla_\theta\mathcal{L}$, any intermediate gradient $\nabla_{f_d}\mathcal{L}$ can be clipped during backpropagation. This can be achieved with a special "*clipping layer*" that behaves like the identity function at the forward pass, and clips the gradient during the backward pass. The resulting cotangeant vector is not a true gradient anymore, but rather a descent direction [47]. In vanilla DP-SGD the clipping is applied on the batched gradient $\nabla_{W_d}\mathcal{L}$ of size $b \times h^2$ for matrix weight $W_d \in \mathbb{R}^{h \times h}$ and clipping this vector can cause memory issues or slowdowns [6]. In our case, $\nabla_{y_D}\mathcal{L}$ is of size $b \times h$ which reduces overhead.

## 3.2 Lip-dp library

To foster and spread accessibility, we provide an opensource tensorflow library for Clipless DP-SGD training, named `lip-dp`. It provides an exposed Keras API for seamless usability. It is implemented as a wrapper over the Lipschitz layers of `deel-lip`[3] library [48]. Its usage is illustrated in Figure 1.

## 4 Experimental results

We validate our implementation with a speed benchmark against competing approaches, and we present the privacy/utility Pareto front that can be obtained with GNP networks.

**Speed and memory consumption.** We benchmarked the median runtime per epoch of vanilla DP-SGD against the one of Clipless DP-SGD, on a CNN architecture and its Lipschitz equivalent respectively. The experiment was run on a GPU with 48GB video memory. We compare against the implementation of `tf_privacy`, `opacus` and `optax`. In order to allow a fair comparison, when evaluating Opacus, we reported the runtime with respect to the logical batch size, while capping the physical batch size to avoid Out Of Memory error (OOM). Although our library does not implement logical batching yet, it is fully compatible with this feature.

An advantage of projection $\Pi$ over per-sample gradient clipping is that the projection cost is

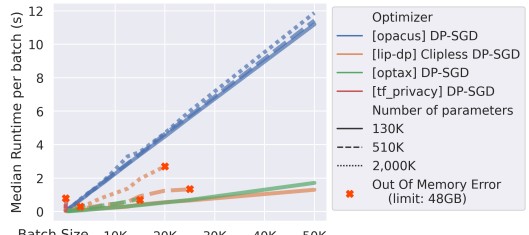

Figure 4: **Our approach outperforms concurrent frameworks in terms of runtime and memory:** we trained CNNs (ranging from 130K to 2M parameters) on CIFAR-10, and report the median batch processing time (including noise, and constraints application $\Pi$ or gradient clipping).

independent of the batch size. Fig 4 validates that our method scales much better than vanilla DP-SGD, and is compatible with large batch sizes. It offers several advantages: firstly, a larger batch size contributes to a decrease of the sensitivity $\Delta \propto 1/b$, which diminishes the ratio between noise and gradient norm. Secondly, as the batch size $b$ increases, the variance decreases at the parametric rate $\mathcal{O}(\sqrt{b})$ (as demonstrated in appendix), aligning with expectations. This observation does not apply to DP-SGD: gradient clipping biases the direction of the average gradient, as noticed by [7].

---

[3]`https://github.com/deel-ai/deel-lip` distributed under MIT License (MIT).

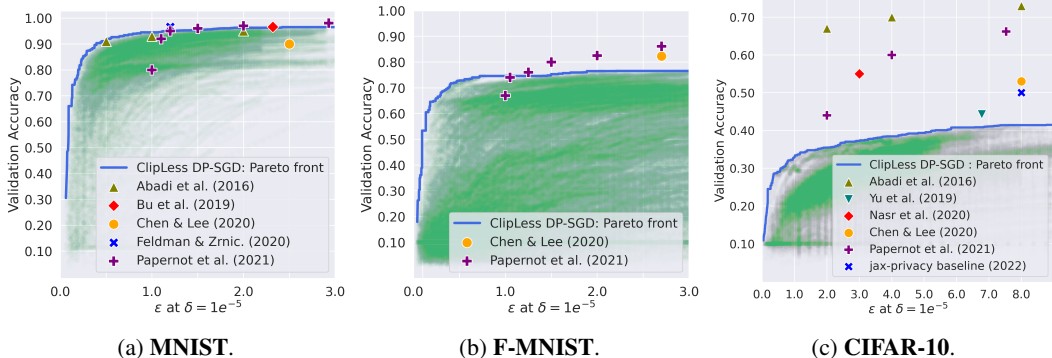

|  (a) **MNIST**.  |  (b) **F-MNIST**.  |  (c) **CIFAR-10**.  |

Figure 5: **Our framework paints a clearer picture of the privacy/utility trade-off.** We trained models in an "out of the box setting" (no pre-training, no data augmentation and no handcrafted features) on multiple tasks. While our results align with the baselines presented in other frameworks, we recognize the importance of domain-specific engineering. In this regard, we find the innovations introduced in [49, 50, 51] and references therein highly relevant. These advancements demonstrate compatibility with our framework and hold potential for future integration.

**Pareto front of privacy/utility trade-off.** We performed a search over a broad range of hyper-parameters values to cover the Pareto front between utility and privacy. Results are reported in Figure 5. We emphasize that our experiments did not use the elements behind the success of most recent papers (pre-training, data preparation, or handcrafted feature are examples). Hence our results are more representative of the typical performance that can be obtained in an "out of the box" setting. Future endeavors or domain-specific engineering can enhance the performance even further, but such improvements currently lie beyond the scope of our work. We also benchmarked architectures inspired from VGG [52], Resnet [53] and MLP_Mixers [54] see appendix for more details. Following standard practices of the community [2], we used *sampling without replacement* at each epoch (by shuffling examples), but we reported $\epsilon$ assuming *Poisson sampling* to benefit from privacy amplification [31]. We also ignore the privacy loss that may be induced by hyper-parameter search, which is a limitation per recent studies [5], but is common practice.

# 5 Limitations and future work

Although this framework offers a novel approach to address differentially private training, it introduces new challenges. We primary rely on GNP networks, where high performing architectures are quite different from the usual CNN architectures. As emphasized in Remark 2, we anticipate that progress in these areas would greatly enhance the effectiveness of our approach. Additionally, to meet requirement 3, we rely on projections, necessitating additional efforts to incorporate recent advancements associated with differentiable reparametrizations [42, 43]. It is worth noting that our methodology is applicable to most layers. Another limitation of our approach is the accurate computation of sensitivity $\Delta$, which is challenging due to the non-associativity of floating-point arithmetic and its impact on numerical stability [55]. This challenge is exacerbated on GPUs, where operations are inherently non-deterministic [56]. Finally, as mentioned in Remark 1, our propagation bound method can be refined.

# 6 Concluding remarks and broader impact

Besides its main focus on differential privacy, our work provides **(1) a motivation to further develop Gradient Norm Preserving architectures**. Furthermore, the development of networks with known Lipschitz constant with respect to parameters is a question of independent interest, **(2) a useful tool for the study of the optimization dynamics** in neural networks. Finally, Lipschitz networks are known to enjoy certificates against adversarial attacks [17, 57], and from generalization guarantees [13], without cost in accuracy [18]. We advocate for the spreading of their use in the context of robust and certifiable learning.

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
