# OpenReview forum: "DP-SGD Without Clipping: The Lipschitz Neural Network Way"
_NeurIPS.cc/2023/Conference — Submitted to NeurIPS 2023_

### Official Review · Reviewer_vJyY · 2023-07-06

**Soundness:** 3 good
**Presentation:** 3 good
**Contribution:** 4 excellent
**Rating:** 5
**Confidence:** 4

**Summary:**

- The paper investigates the use of Lipschitz constrained networks to replace clipping functions and limit gradient sensitivity in DP-SGD.
- Lipschitz constrained networks are utilized as an alternative to clipping in order to address the issues of clipping's impact on convergence and performance in DP-SGD.


**Strengths:**

- The idea of removing clipping as an alternative to clipping itself is promising, as clipping is known to have detrimental effects on convergence and performance of DP-SGD, even without noise addition [1].
- The paper introduces the replacement of Vector-Jacobian product with Scalar-Scalar product to reduce computational complexity. The proposed methods outperform existing SGD approaches by a significant margin in terms of speed, which is crucial as memory usage and time inefficiency are major drawbacks of DP-SGD.

[1] Differntially Private Shaprness-Aware Training (ICML’23)

**Weaknesses:**

- Please refer to the questions.
- (Minor) There are several typos, such as the use of "cotangeant vector" which sounds little awkward, and inconsistencies in figure references (e.g., Fig 4 vs. Figure 5). Please carefully review the grammar and correct the typos.

**Questions:**

- The main concern is whether "Clipless DP-SGD" is truly effective compared to "Clip DP-SGD" in terms of training stability and performance. While it is understood that the main idea is to address the detrimental effects of clipping by proposing a clipping-free approach, the proposed method still relies on approximations or has drawbacks such as the Lipschitz constrained condition or gradient approximation with scalar-scalar form. Thus, (i) Can you provide mathematical bounds to support the strength of your methods? or (ii) Can you provide additional experimental results beyond Figure 5 to further demonstrate the performance of your approach?
- Recent papers have explored different ways of using DP-SGD, particularly in fine-tuning tasks, to mitigate memory usage and time inefficiency [2,3]. Considering that their approaches may differ from training from scratch tasks, it would be worth to investigate whether your proposed networks are well-suited for fine-tuning tasks to improve generalization performance.
- Can you provide more details on the implementation of GNP networks in DP-SGD? The implementation of 1-Lipschitz seems to be naive and detrimental to traininig, similar to selecting a norm clipping value in clip DP-SGD. However, the authors mention a problem in this implementation, stating, "With Gradient Norm Preserving (GNP) networks, we expect to mitigate this issue" (line number 236). Could you further explain how your methods address this problem? or let me know that my understanding is incorrect.

I’d be happy to raise my score if the authors can address the weaknesses/questions in the rebuttal.

[2] Large language models can be strong differentially private learners (ICLR’22)

[3] Scalable and efficient training of large convolutional neural networks with differential
privacy (NeurIPS’22)

**Limitations:**

The paper provides a detailed discussion of its limitations.

---

> ### Author Rebuttal · Authors · 2023-08-08
>
> Thank you for your careful reading and your challenging questions.
>
> #### **Question (i)**
>
> Note that the scalar-scalar form is not an approximation: it is a valid upper bound (the “=” symbol is a typo, it is actually “<=”) coming from applying Cauchy-Swartz inequality to a matrix-vector product. Therefore all the guarantees reported are correct, and that has been verified empirically. Could you detail the nature of the mathematical bounds you wish to see? If it concerns the expressiveness of Lipschitz constrained networks, please see our common response. For more details on the computation of gradient bounds, and on the variance of the gradient, you can take a look at the proofs in Appendix E.2 and E.3. For an analysis on the consequence of gradient clipping in Lipschitz networks see Appendix B.5.
>
> #### **Question (ii)**
>
> We ran additional experiments on two tabular datasets (Adult and Breast Cancer) , where our approach is competitive with vanilla DP-SGD for a comparable hyperparameter optimization budget. Please see the general response and attached PDF for detailed results.
>
> > Can you provide more details on the implementation of GNP networks in DP-SGD
>
> The whole network can be constrained to be 1-Lipschitz with respect to the input, and will remain able to fit arbitrary decision boundaries (see Bethune et al.) so there is no need to tune the Lipschitz constant.
> Moreover the parametrization is not so naive, as discussed in Appendix B.1. In short, we rely on spectral normalization to enforce the 1-Lipschitz constraint. The leading eigenvector (corresponding to maximum singular value) is cached from one gradient step to another to speed-up computations. Moreover, Bjorck algorithm (see Anil et al.(2019)) projects matrices on the Stiefel manifold (i.e close to being orthogonal), which ensures that the gradient norm remains unchanged after one gradient step. Indeed, if the matrix is orthogonal, then the Jacobian is orthogonal too, which ensures that VJP products are norm preserving (see Li et al (2019)).
>
> *Béthune, L., Boissin, T., Serrurier, M., Mamalet, F., Friedrich, C. and Gonzalez Sanz, A., 2022. Pay attention to your loss: understanding misconceptions about lipschitz neural networks. Advances in Neural Information Processing Systems, 35, pp.20077-20091.*
>
> *Anil, C., Lucas, J. and Grosse, R., 2019, May. Sorting out Lipschitz function approximation. In International Conference on Machine Learning (pp. 291-301). PMLR.*
>
> *Li, Q., Haque, S., Anil, C., Lucas, J., Grosse, R.B. and Jacobsen, J.H., 2019. Preventing gradient attenuation in lipschitz constrained convolutional networks. Advances in neural information processing systems, 32.*
>
> > it would be worth to investigate whether your proposed networks are well-suited for fine-tuning tasks to improve generalization performance
>
> Although this paper focuses on training networks from scratch, it is worth noting that it is compliant with fine tuning in two different ways:
> 1) By using a 1-lipschitz backbone (recent work of Serrurier et al. trained a backbone that reached 70% accuracy on imagenet 1k)
> 2) Our algorithm is compliant with the partial training of the last layers: one can finetune a standard backbone with the last layers replaced by their Lipschitz equivalent. The only non-trivial change is that the input of this last layer must be bounded, which can be enforced easily.
>
> We leave this for future work.

---

> > ### Comment · Reviewer_vJyY · 2023-08-17
> >
> > Thank you for responses and clarifications that the authors provided.
> >
> > - One of my remaining questions is about the performance aspect. As my first question is little ambiguous, I point out that my original question point was to highlight that GNP appears to be inherently more effective than gradient clipping in the context of deep learning optimization even though theoretically the proposed methods have smaller upper bound. My concern arose after observing the CIFAR-10 accuracies depicted in Figure 5. Although the authors mention the potential for future integration, it seems, in my view, that employing GNP is still worried rather than using gradient clipping. At least in my opinion, using GNP is better than gradient clipping when $\epsilon$ is small.

---

> > > ### Author Response · Authors · 2023-08-17
> > >
> > > Thank you for your insighful question.
> > >
> > > >  At least in my opinion, using GNP is better than gradient clipping when is small.
> > >
> > > Your remark raises interesting questions about the dynamics of DP training in neural networks. This reminds us the following excerpt of De et al (p6 of their work) that we’d like to share here:
> > >
> > > "*[...] reducing the variance introduced by noise may be more important than reducing the bias introduced by clipping.*"
> > >
> > > We remind their concern was attaining high accuracy in higher $\epsilon$ regimes.
> > >
> > > Gradient clippings reduces variance at the cost of increased bias. Clipless DP-SGD removes the bias but suffers from higher variance if the gradient bounds are too lose. If your hypothese is correct, it means that low bias is more important is low $\epsilon$ regimes, and reduced is more important is higher $\epsilon$ regimes. Intuitively, that seems natural.
> > >
> > > We tested your hypothesis by tuning hyper-parameters in the small $\epsilon$ regime, and we obtain the following results on Cifar-10:
> > >
> > > |  **Epsilon**  | 3.5 | 4.1 | 4.9 |
> > > | ---- |  ----- | ---- | ----- |
> > > | **Val. Acc.**	|   38.9  | 40.4 | 42.1  |
> > >
> > > With smaller epsilon constraints:
> > >
> > > |  **Epsilon**  | 0.75 | 0.96 | 1.14
> > > | ---- |  ----- | ---- | --- |
> > > | **Val. Acc.**	|   31.5  | 34.2 | 35.2

---

### Official Review · Reviewer_Coyf · 2023-07-06

**Soundness:** 2 fair
**Presentation:** 3 good
**Contribution:** 2 fair
**Rating:** 4
**Confidence:** 3

**Summary:**

Differentially Private (DP) Deep Neural Networks (DNNs) face challenges in estimating tight bounds on the sensitivity of the network’s layers. Instead, they rely on a per-sample gradient clipping process (as argued by the authors). This process not only biases the direction of the gradients but also proves costly in both memory consumption and computation. To provide sensitivity bounds and avoid the drawbacks of the clipping process, the authors provide a theoretical analysis of Lipschitz constrained networks, and uncovers a previously unexplored link between the Lipschitz constant with respect to their input and the one with respect to their parameters. By bounding the Lipschitz constant of each layer with respect to its parameters, the authors argue it will guarantee DP training of these networks.

**Strengths:**

The paper is well-structured and clearly written.

The theoretical part is simple and easy to follow.

**Weaknesses:**

Estimating Lipschitzness with respect to parameters may not be necessary. If the network is Lipschitz continuous with respect to the input, its gradient will be bounded, and thus the weight update will also be bounded. So, the motivation may not be rational.

Experimental results do not support the arguments. The validation accuracy of the DP-SGD is lower than several referenced works.



**Questions:**


1. Adam uses a normalized gradient between the first-order momentum and square root of the second-order momentum, which makes it robust to Lipschitz variations in different layers. Does it still need gradient clipping?

2. Regarding per-sample gradient clipping, in the current deep learning framework, we usually do not use per-sample gradient clipping. Instead, we apply gradient clipping after obtaining the averaged gradients. In the abstract, the authors mention the difficulty of per-gradient clipping, but I don't understand the reason?

3. Also, gradient clipping is a fast operation. I don't understand why it results in higher memory consumption and computation.

4. Regarding the experiments, the results do not show that the DP-SGD achieves better performance (validation accuracy) than the referenced works. How should we evaluate the effectiveness of this work?





**Limitations:**

See weaknesses and my questions

---

> ### Author Rebuttal · Authors · 2023-08-08
>
> Thank you for your efforts in engaging in this discussion. From what we understand, there are a few points which we need to clarify further.
> As for the summary, our main contribution is to make the DP training process quicker and more memory efficient by **eliminating** the need for per-sample gradient clipping. Our new algorithm is named **Clipless DP-SGD**, while the “DP-SGD” you refer to is the seminal work of Abadi et al (2016).
>
> > in the current deep learning framework, we usually do not use per-sample gradient clipping. Instead, we apply gradient clipping after obtaining the averaged gradients. [...] In the abstract, the authors mention the difficulty of per-gradient clipping, but I don't understand the reason?
>
> There is a common confusion between the *per-sample gradient* clipping of DP training literature with the one used in non-private training (where people usually work on the *averaged gradient*).
>
> In Differential Privacy, the noise must be calibrated to the sensitivity of the function (here, the gradient computation) to the change of one data point (see the seminal paper of the field of Differential Privacy, Dwork et al (2006)). For a traditional neural network it is not possible to compute bounds on the sensitivity for parameter-wise gradient (which is the source of the privacy leakage), as observed in Abadi et al (2016). This is why they proposed DP-SGD: a variation of SGD in which the per-sample gradients are clipped, which ensures they are bounded. This idea is at the core of all DP libraries for deep learning (Opacus, tf_privacy, jax_privacy, etc.), except ours. In order to grasp the difficulty of implementing per-sample clipping, please take a look at this implementation of per-sample gradient computations in Linear layers of Opacus library: [it can be cumbersome and costly](https://github.com/pytorch/opacus/blob/main/opacus/grad_sample/linear.py).
>
> *Abadi, M., Chu, A., Goodfellow, I., McMahan, H.B., Mironov, I., Talwar, K. and Zhang, L., 2016, October. Deep learning with differential privacy. In Proceedings of the 2016 ACM SIGSAC conference on computer and communications security (pp. 308-318).*
>
> *Dwork, C., McSherry, F., Nissim, K. and Smith, A., 2006. Calibrating noise to sensitivity in private data analysis. In Theory of Cryptography: Third Theory of Cryptography Conference, TCC 2006, New York, NY, USA, March 4-7, 2006. Proceedings 3 (pp. 265-284). Springer Berlin Heidelberg.*
>
> > If the network is Lipschitz continuous with respect to the input, its gradient will be bounded, and thus the weight update will also be bounded [...] Estimating Lipschitzness with respect to parameters may not be necessary.
>
> Your intuition is correct but this requires additional hypotheses (like Lipschitz loss, bounded input, bounded biases): our whole framework is about introducing those necessary hypotheses (Section E in appendix) and automating the computation of the bound. Without those hypotheses, a conventional neural network **is not** Lipschitz with respect to its parameters over the whole domain.
> Knowing that the gradient is bounded is not sufficient: to report valid $(\epsilon,\delta)$ guarantees we **need** to calibrate the noise, which requires to know in advance the exact upper bounds on the gradient. Without this knowledge, no DP guarantees can be given.
>
> > Also, gradient clipping is a fast operation. I don't understand why it results in higher memory consumption and computation.
>
> With private training, we have to clip the gradients on a per-sample basis, which significantly increases runtime and memory consumption. The clipping itself is not the most expensive operation, but rather the cost comes from the computation of per-example gradients. This has been observed since the early work of Abadi et al (2016), and has proven to remain a difficult challenge to this day (please read this excellent survey “How to DP-fy ML?” and references therein for more details). It can also be seen on the runtime of competing frameworks (Figure 4 of our paper). While efficient implementations exist, like discussed in the early work of Goodfellow (2015) or a recent paper of Lee and Kifer (2021), it has proven challenging to overcome for frameworks like tensorflow or pytorch. To the best of our knowledge, only Jax natively supports these operations efficiently.
>
> *Goodfellow, I., 2015. Efficient per-example gradient computations. arXiv preprint arXiv:1510.01799.*
>
> *Ponomareva, N., Hazimeh, H., Kurakin, A., Xu, Z., Denison, C., McMahan, H.B., Vassilvitskii, S., Chien, S. and Thakurta, A.G., 2023. How to dp-fy ml: A practical guide to machine learning with differential privacy. Journal of Artificial Intelligence Research, 77, pp.1113-1201.*
>
> *Lee, J. and Kifer, D., 2021. Scaling up differentially private deep learning with fast per-example gradient clipping. Proceedings on Privacy Enhancing Technologies, 2021(1).*
>
> > Adam [...]  Does it still need gradient clipping?
>
> Yes! For example, [tf_privacy has its own implementation](https://github.com/tensorflow/privacy/blob/fafa69b65c2a20c8e06cab7032e8b25e8dbcea43/tensorflow_privacy/privacy/optimizers/dp_optimizer_keras.py#L600).
>
> > Experimental results do not support the arguments
>
> This is not true. There are no overclaims in the paper. We claimed that our framework was among the fastest, and this can be verified in Figure 4. We did not claim that our framework yielded SOTA results in Figure 5. Hence, the experimental results support the claims.
>
> Once again, thank you for engaging in the discussion. We will use your feedback to improve the introduction and the context. Please do not hesitate to tell us which sections and parts of the paper you would like further clarifications on.

---

> > ### Comment · Reviewer_Coyf · 2023-08-18
> > **Official Comment by Reviewer Coyf**
> >
> > Thank you for your responses and clarifications.
> >
> > - I remain unconvinced by the argument that per-sample gradient clipping significantly enhances speed. Consider the computational complexity of a convolution operator, which stands at $O(C_{in} \times C_{out} \times K \times K \times W \times H)$. In contrast, gradient clipping necessitates only O($C_{in} \times C_{out} \times K \times K$) comparison operators. Similarly, for a linear layer, the computational complexity is $O(C_{in} \times C_{out} \times D)$, while gradient clipping requires only O($C_{in} \times C_{out}$) comparison operators. In both convolutional and linear layers, the reduction in computation appears negligible.
> >
> > - Numerous studies delve into the realm of local or global Lipschitz continuity. I remain skeptical about the challenge in guaranteeing bounded inputs, loss, and bias terms.
> >
> > Thus, I prefer to keep my previous rating.

---

> > > ### Author Response · Authors · 2023-08-18
> > >
> > > > I remain unconvinced by the argument that per-sample gradient clipping significantly enhances speed
> > >
> > > I think there is a misunderstanding here: **per-sample gradient clipping is the solution of litterature, not the one of our paper**. Beside, as we explain in our review, the computional cost comes from per-sample gradient computation.
> > >
> > > We are confused by your review. It feels like your criticizing the whole field of research rather than our contribution. You say you remain unconvinced, but did you read the references we sent ? If you believe that speed is not an issue in DP training, are you suggesting that existing libraries like Opacus or tf_privacy are not implemented correctly ? Are you suggesting that the survey "*How to dp-fy ml*" (that was presented in ICML 2023 as a tutorial) is ill-informed on the challenges of the field ?
> > >
> > > How do you explain the speed improvement of our framework over concurrent frameworks if you think the problem does not exist at all ?
> > >
> > > > Similarly, for a linear layer, the computational complexity is [...] convolutional and linear layers, the reduction in computation appears negligible.
> > >
> > > You omitted to take into account the batch size $B$ in your complexity evaluation. Your computations are only valid for gradient clipping on the *average gradient*, whereas the field of DP training relies on *per-sample gradient* computations. Therefore, it is not relevant for DP training. We suggest you take a look at Section 4 of the seminal paper of Abadi et al. We hope their analysis can convince you.
> > >
> > > > Numerous studies delve into the realm of local or global Lipschitz continuity. I remain skeptical about the challenge in guaranteeing bounded inputs, loss, and bias terms.
> > >
> > > Would you mind citing a *single one* of those "numerous studies" that provides formal Lipschitz guarantees over parameter space like we do ? Can you show us a *single neural network* (beside ours) for which you know the global upper bound of the Lipschitz constant over the parameter space ?

---

### Official Review · Reviewer_mBgX · 2023-07-06

**Soundness:** 3 good
**Presentation:** 4 excellent
**Contribution:** 2 fair
**Rating:** 3
**Confidence:** 4

**Summary:**

The paper addresses the problem of efficiently bounding the sensitivity of gradients in DP-SGD by using special architectures the layers of which can be proven to be Lipschitz with respect to the parameters, hence bounded gradient. They show how to recursively calculate the sensitivity of a sequence of layers, and incorporate the method in an algorithm to perform private SGD without clipping.

**Strengths:**

The writing is exceptionally lucid. The concept is original and potentially significant, although there are at present many limitations.

**Weaknesses:**

There are a lot of constraints on the architecture that severely limit the potential of the method for short-term impact. I'm torn, because introducing the concept at this stage is of value, but far more work must be done -- both theoretical, in establishing the requisite bounds for popular architectures -- and experimental, in demonstrating that the approach achieves good points on the privacy/utility/efficiency Pareto frontier -- before we can assess the significance of the work.

I'm not convinced that it isn't a major problem that the gradients can vanish during training. This is the reason for the success of adaptive (layer-wise) clipping strategies. In particular see "EXPLORING THE LIMITS OF DIFFERENTIALLY PRIVATE DEEP LEARNING WITH GROUP-WISE CLIPPING" which would seem to enjoy the efficiency of your approach without the drawbacks of vanishing gradients or restricted architecture class.

**Questions:**

What is meant by “the activation is centered in zero”-- in particular, is ReLU really centered?

Experimental results. Figure 4 is too small. I can't see tensorflow_privacy? Also it appears that the advantage over optax is negligible. In Figure 5, the other algorithms get one or a few points, while their own algorithm is given the benefit of thousands of hyperparameter choices to establish the Pareto frontier. Moreover the result is negative: for EMNIST the other algorithms match yours, and for the other datasets the other algorithms have higher accuracy.

Probably the DP definition should be using add/remove-one neighborhoods, in accordance with the use of (pretend) Poisson sampling for accounting.

**Limitations:**

Limitations are honestly and adequately discussed. No potential negative societal impact.

---

> ### Author Rebuttal · Authors · 2023-08-08
>
> Despite your rating, we note that you recognize the originality of our work and its potential impact, and that you did not find any major issue in our method nor our methodology.
>
> Your concern seems to be about the short term impact of the paper. We are not convinced that beating SOTA results is necessary for a novel idea to be interesting to the scientific community, especially when it is accompanied by a Python package to ease reproducibility and future evaluations. Below, we detail why the theoretical and empirical constraints you pointed out are not so severe.
>
> ### Theoretical work
>
> > There are a lot of constraints on the architecture that severely limit the potential of the method for short-term impact.
>
> The `DP_Layer` blocks we implemented are flexible enough to accommodate most popular neural network architectures used in computer vision or tabular data (e.g convolutions, Lipschitz pooling, Lipschitz activations, layer centering in Appendix C.2.3, etc). These blocks also allow residual connections. For instance, we easily re-implemented Lipschitz constrained ResNets (see Appendix B4 p17, and Figure 8 p18), VGGs and even the recent MLP Mixers [54] (Appendix C.2.4), and we report their individual results in Figure 10 p26. You can also take a look at the folder “experiments” in the Python package, or the documentation of the Python package. Note that Theorem 1 is also quite general and covers most feed-forward architectures.
>
> See our common response for more details about the expressiveness of this family of architectures.
>
> ### Experimental work
>
> Please see our general response regarding the positioning of the paper,the empirical comparison to prior work, and additional experiments against a vanilla DP-SGD baseline.
>
> In Lipschitz networks training, vanishing gradients can be a serious issue (see Li et al (2019)), and they yield unfavorable signal to noise ratio during backprop for private training with our clipless method. We are grateful for your suggestion regarding the use of group-wise gradient clipping. Nonetheless, the method you mention does not intend to get rid of per-sample clipping, hence it is orthogonal to ours. The two methods could however be complementary and their combination investigated in future work.
>
> *Li, Q., Haque, S., Anil, C., Lucas, J., Grosse, R.B. and Jacobsen, J.H., 2019. Preventing gradient attenuation in lipschitz constrained convolutional networks. Advances in neural information processing systems, 32.*
>
> ### Questions
>
> > What is meant by “the activation is centered in zero”-- in particular, is ReLU really centered?
>
> Activation is centered in zero means that $\sigma(0) = 0$. We define it formally in Assumption 2 p28 in the appendix. This includes ReLu, LeakyRelu, tanh, Groupsort, and a few others. This hypothesis allows tighter bounds in Theorem 1 than just assuming “bounded activations”. We will clarify the meaning of this hypothesis.
>
> > Figure 4 is too small. I can't see tensorflow_privacy?
>
> Thank you for your comment. We will make the figure bigger.  The tensorflow_privacy library seems inefficient in terms of runtime and memory usage, therefore we had to cap the plot to batch sizes not bigger than 256.
>
> > In Figure 5, the other algorithms get one or a few points, while their own algorithm is given the benefit of thousands of hyperparameter choices to establish the Pareto frontier
>
> The results we give as baselines for the traditional DP-SGD are results specifically selected by previous papers that typically required extensive tuning across multiple studies to yield SOTA results on these datasets. These papers often do not provide a comprehensive figure of the Pareto frontier and our aim was to be more transparent about our privacy/utility values on a larger spectrum of privacy budgets. Also the green lines you see appear since we log the (epsilon,val_accuracy) tuple corresponding to every epoch of every run: an individual point is not yielded by one hyperparameter choice, rather, it is the status of one of our experiments at one point in time. Thanks to the efficiency of our framework, these experiments have been conducted on a single GPU in a few days. We see this exhaustivity as further proof that our method scales well.  *Since you are not the only reviewer to have misinterpreted this figure we will modify the caption.*
>
>
> > Probably the DP definition should be using add/remove-one neighborhoods
>
> Your remark is correct, we will modify the definition accordingly.

---

> > ### Comment · Reviewer_mBgX · 2023-08-14
> > **Response to rebuttal**
> >
> > I appreciate that it should not be necessary to beat all SOTA results as a precondition for initial publication. But I feel there is a high potential for this method to produce poor models in terms of accuracy, and there are not sufficient experimental results to demonstrate that this is not occurring. My concern is the same as reviewer 6vik's weakness #1. Without clipping, if you have very small gradient norms, your signal-to-noise ratio is low, and the trained model will have poor performance. Clipping lets you control this signal-to-noise ratio more effectively, particularly wtih adaptive clipping, or better yet layer-wise adaptive clipping, which also solves the efficiency problems around per-example clipping. So I feel there needs to be more clear empirical evidence that the method does not suffer compared to those methods in terms of accuracy.
> >
> > Another idea that might help to alleviate this concern would be to actually measure the empirical gradients and report their distribution: what is the empirical signal-to-noise ratio? But that would be outside of the scope of this review process.
> >
> > I maintain that it is not fair to show your Pareto frontier compared to a single point from another algorithm. I see your point that ideally we would all be publishing our Pareto frontiers, but since we are not, the honest comparison is to run your own hyperparameter tuning and to compare it to the other published result.
> >
> > Thank you for clarifying that the architectural constraints are not as severe as I had thought. I will increase my score to 3 and decrease my confidence.

---

> > > ### Author Response · Authors · 2023-08-17
> > >
> > > Thank you for opening this discussion.
> > >
> > > > Another idea that might help to alleviate this concern would be to actually measure the empirical gradients and report their distribution
> > >
> > > We did it in **figures 6.b) and 7.b) in appendix p17**. We see that the gradient norms for GNP networks is *higher* than the one of conventional networks.
> > >
> > > > Without clipping, if you have very small gradient norms, your signal-to-noise ratio is low, and the trained model will have poor performance.
> > >
> > > The signal-to-noise ratio is not the only metric that influences the final utility: the gradient need not to be unbiased for the signal to be actually useful. By getting rid of clipping, we also get rid of the induced bias.
> > >
> > > > I appreciate that it should not be necessary to beat all SOTA results as a precondition for initial publication [...] I see your point that ideally we would all be publishing our Pareto frontiers
> > >
> > > We agree; that is indeed a fairer experimental procedure, and our work pushes towards this ideal.
> > >
> > > > I feel there is a high potential for this method to produce poor models in terms of accuracy [...]
> > >
> > > Note that in the rebuttal we provided 2 new datasets on which the final utility is competitive (or exceed) the one of vanilla DP-SGD.
> > >
> > > > I maintain that it is not fair to show your Pareto frontier compared to a single point from another algorithm [...] the honest comparison is to run your own hyperparameter tuning and to compare it to the other published result
> > >
> > > When offering a library, average behavior becomes more indicative of typical performance than seeking out the extremes. In a scenario with an unlimited computational budget, even a random baseline holds a nonzero chance of outperforming the current state of the art (SOTA). Hence, when we emphasize the highest score in our reporting, what are we truly measuring apart from the computational budget?
> > >
> > > However, we understand your overall concern, please see this additional experiment of Clipless DP-SGD that uses standardized Cifar-10 dataset (assuming no privacy loss in computing mean and std, just like Opacus tutorial):
> > >
> > > | **Epsilon** | 4.05 | 5.0 | 8.01 | 11.1 | 15.03 |
> > > | ----- | ------ | ------ | ------ | ----- | ---- |
> > > | **Val. Acc.**| 37.6 | 39.3 | 42.0 | 43.2 | 44.5 |
> > >
> > > We see that we close the gap with the results of De et al, which are among the top-performers on Cifar-10.

---

> > > > ### Comment · Reviewer_mBgX · 2023-08-17
> > > >
> > > > Thank you for pointing me to the gradient norm distribution figures in the appendix. In figures 6/7.b) what are the parameters for which this gradient norm distribution is computed? Random or trained? It is stated that the norms are "sticking closer to the upper bound we are able to compute for the gradient norm" -- I can see that the gradients are larger for the GNP network, but what is that norm bound?
> > > >
> > > > Sorry, I cannot find the work you are comparing to with these new experiments, De et al. Can you point me to the exact table of values I should compare your accuracies with?

---

> > > > > ### Author Response · Authors · 2023-08-18
> > > > >
> > > > > Thank you for your engagement in our work. We appreciate your thoroughness.
> > > > >
> > > > > In Fig 6. and 7. the report the bounds for Dense layers in a MLP of depth 3
> > > > >
> > > > > > Random or trained?
> > > > >
> > > > > It is computed on the whole dataset at initialization.
> > > > >
> > > > > > In figures 6/7.b) what are the parameters for which this gradient norm distribution is computed?
> > > > >
> > > > > In fig 6., we agregate the individual bounds on the layers to obtain a global bound on the whole gradient vector, whereas in fig 7. we plot the distributions of individuals layers.
> > > > >
> > > > > >  but what is that norm bound?
> > > > >
> > > > > In fig 6.b), the global upper bound is 135.66. Less than two times the maximum empirical bound of 73.
> > > > >
> > > > > In fig 7.b) the common upper bound of each of the gradient bound (for every layer) is 78.38. Not too far away from the maximum empirical bound of 56.
> > > > >
> > > > > For the conventional network without Lipschitz constraints we don't have analytical upper bounds.
> > > > >
> > > > > We will add these informations to the figure.
> > > > >
> > > > > > Sorry, I cannot find the work you are comparing to with these new experiments, De et al. Can you point me to the exact table of values I should compare your accuracies with?
> > > > >
> > > > > The closest one is the first row of their table 2 (page 7) where their model is a Wide-Resnet, and where they rely on the same pre-processing.

---

> > > > > > ### Comment · Reviewer_mBgX · 2023-08-18
> > > > > >
> > > > > > So fig 7b shows the distribution of the gradient norms of a GNP at initialization (randomly initialized parameters). What about at other points in training? Would you expect the gradient norm distribution to shift toward zero as the model converges? For a more clear picture, we would need to see this gradient norm distribution during the main phase of learning when losses are dropping, and again at convergence.
> > > > > >
> > > > > > Thank you for sharing the upper bounds, but to be honest I am disappointed that I had to ask for them: since you referred to it ("sticking close to the upper bound") it is misleading that you didn't already write it in the paper so the reader can judge for themselves whether it is close or not. Now you want to compare the maximum gradient norm at initialization to the norm bound to argue you have a good signal-to-noise ratio. But the mean/median/mode looks more like 20-30 in 7b, or about 3-4x smaller than the bound. I appreciate your point that bias from clipping is also worth considering, but having noise on a scale of 3-4x the typical gradient will surely degrade performance significantly.
> > > > > >
> > > > > > Now you claim to "close the gap" to De et al., where their most basic model with eps=8 has a validation accuracy of 50.8 compared to your least private model (eps=15) with 44.5.
> > > > > >
> > > > > > I'm sorry, I actually really want to support this work: as I said the writing is very clear, the idea is interesting, the experiments are extensive and it's wonderful that you published the library, but in the end I am simply not convinced that the method will actually be useful to anyone in practice due to this issue of having to add so much noise relative to the gradient norms. Clipping may have its downsides, but it is simple and relatively effective, particularly with (layer-wise) adaptive clipping.

---

> > > > > > > ### Author Response · Authors · 2023-08-19
> > > > > > >
> > > > > > > >  I actually really want to support this work: as I said the writing is very clear, the idea is interesting, the experiments are extensive and it's wonderful that you published the library
> > > > > > >
> > > > > > > Thank you for your encouraging words. Increasing your rating is the best support for our work you can provide here. We understand your motivations for rejection, but we disagree on the substance.
> > > > > > >
> > > > > > > The peer-review process is integral to refining and validating our approach. We are encouraged by the positive responses from all reviewers, including yourself, regarding the validity of our method.
> > > > > > >
> > > > > > > The 'lipdp' package serves multiple purposes within our work. It not only facilitates reproducibility but also extends a valuable resource to the community. By offering 'lipdp,' we aim to streamline the evaluation of enhancements built upon "vanilla" Clipless-DP-SGD.  It is important to note that Clipless-DP-SGD is a standalone contribution, deserving its own dedicated focus. Integrating other methods within the same paper might compromise clarity and lead to potential misinterpretations of the true contribution.
> > > > > > >
> > > > > > > > Would you expect the gradient norm distribution to shift toward zero as the model converges
> > > > > > >
> > > > > > > Indeed, the gradient of the loss w.r.t the logits $\nabla_{y}\mathcal{L}$ tends toward zero when the accuracy is high: reaching high accuracy is exactly what we want. And if the accuracy is not high, then the gradient norm remains high too. This quantity is a common upper-bound of all other gradient bounds. The same phenomenon occurs in vanilla DP-SGD: at some point in training the noise overrun the signal when gradients are smaller than the clipping value. We will add the corresponding figure showcasing the evolution of the histograms.
> > > > > > >
> > > > > > > > I appreciate your point that bias from clipping is also worth considering, but having noise on a scale of 3-4x the typical gradient will surely degrade performance significantly.
> > > > > > >
> > > > > > > You raise a valid concern, but you are also missing an advantage of our method: by getting rid of per-sample gradient computations, we allow higher batch size more easily. We managed a physical batch size of *50,000 examples* in figure 4. Only Opacus managed similar performance, *with virtual batching*. This higher batch size increases the signal-to-noise ratio in proportion.
> > > > > > >
> > > > > > > > it is misleading that you didn't already write it in the paper
> > > > > > >
> > > > > > > We agree it should be part of the figure, we modified the paper in consequence.

---

> > > > > > > > ### Comment · Reviewer_mBgX · 2023-08-19
> > > > > > > >
> > > > > > > > So, toward convergence, the gradient norms become small, while your norm bounds remain constant, decreasing signal-to-noise, is that right? Yes, the same happens with vanilla DP-SGD, but with adaptive clipping, the clipping norm decreases in tandem with the gradient norms, so the signal-to-noise is approximately constant.

---

> > > > > > > > > ### Author Response · Authors · 2023-08-21
> > > > > > > > >
> > > > > > > > > > but with adaptive clipping, the clipping norm decreases in tandem with the gradient norms, so the signal-to-noise is approximately constant
> > > > > > > > >
> > > > > > > > > Adaptive clipping consumes a part of the privacy budget to compute adapt the clipping value, and it introduces new hyper-parameters to the method. Arguably, it is more complex:
> > > > > > > > >
> > > > > > > > > "*One possibility is to use adaptive clipping instead of fixing the
> > > > > > > > > clipping norm a priori Andrew et al. (2021). However, its implementation is more complicated than
> > > > > > > > > static clipping, and thorough tuning of hyperparameters with static clipping norm usually results in
> > > > > > > > > the same utility as adaptive clipping Andrew et al. (2021).*"
> > > > > > > > >
> > > > > > > > > **Source**: Ponomareva, N., Hazimeh, H., Kurakin, A., Xu, Z., Denison, C., McMahan, H.B., Vassilvitskii, S., Chien, S. and Thakurta, A.G., 2023. *How to dp-fy ml: A practical guide to machine learning with differential privacy.* Journal of Artificial Intelligence Research, 77, pp.1113-1201.
> > > > > > > > >
> > > > > > > > > Therefore, we don't think that adaptive clipping is a sufficient argument on itself to rule-out the relevance of our work.
> > > > > > > > >
> > > > > > > > > Finally, note that our framework **is compatible with adaptive clipping**: the clipping layer that clips $\nabla_y\mathcal{L}$ can benefit from this innovation. We did experiments in this direction, and it improved our final accuracy. However, this is outside the scope of Clipless DP-SGD framework to present this, and is more appropriate as a future work.

---

> > > > > > > > > > ### Comment · Reviewer_mBgX · 2023-08-21
> > > > > > > > > >
> > > > > > > > > > The authors are simply not offering credible arguments. Adaptive clipping consumes around 0.5% of the noise budget, and works well out of the box without hyperparameter tuning [1]. It may be that static clipping performs about as well, but as Ponomareva et al. say, "thorough tuning of hyperparameters" is necessary. Actually, in the original adaptive clipping paper [1], adaptive clipping performs better than fixed clipping in three out of six datasets even after tuning the fixed clipping norm, so the DP-ify paper may have mischaracterized it.
> > > > > > > > > >
> > > > > > > > > > I will raise my confidence back to 4.
> > > > > > > > > >
> > > > > > > > > > [1] Andrew et al. (2021) "Differentially Private Learning with Adaptive Clipping"

---

### Official Review · Reviewer_6vik · 2023-07-12

**Soundness:** 3 good
**Presentation:** 3 good
**Contribution:** 3 good
**Rating:** 6
**Confidence:** 3

**Summary:**

The paper studies the question of how to do differentially private optimization without using per-sample gradient clipping, in order to simplify and speedup the iteration cost.
The paper proposes to restrict the class of functions to feed-forward neural networks for which it is feasible to compute bound on the gradient norm (Lipshitz constant), and proposes to compute adaptively the bound on the gradient norm (layer-wise) at every step of DP-SGD depending on the current iterate point. Paper provides the description of the algorithm, as well as evaluates its practical behavior.

**Strengths:**

- An efficient implementation of the algorithm is provided.
- Experiments show that per-iteration runtime of the proposed algorithm is indeed faster.
- Overall the paper is interesting and novel and provides a new direction for future research.

**Weaknesses:**

1. No clear comparison of the proposed algorithm to the baseline method (DP-SGD) is given in terms of the final accuracy. When restricting to the same architecture, it is unclear if the proposed algorithm can still reach the good accuracy compared to the classical DP-SGD with gradient clipping. Without clipping the gradients, the amount of the added DP noise to each gradient is larger than if you clip the gradients, which might hurt the final performance.
2. From the experiments on CIFAR10 one might conclude that for the same privacy $\epsilon$ the final accuracy of the baselines is much better than of the proposed algorithm, which makes the proposed algorithm not applicable.
3. In the “local” strategy (line 201), how exactly did you calculate the amount of the noise to be added? I did not find a clear description of the “local” strategy, and how it is different from the “global” strategy.
4. Some parts of the paper are not very clearly written (see questions below).

**Questions:**

1. What do the green lines in pareto-front on Figure 5 represent ?
2. Remark 1 says that the paper proposes a more efficient way to compute X_d, however I did not find the algorithm in the paper.
3. It is unclear from the current presentation what is the connection of Theorem 1 with Algorithm 1, is it only for the easier computing of step 6 of Algorithm 1 ? What about steps 2 and 7?
4. I did not understand the paragraph in lines 258-267. However some hybrid approach might be indeed a good solution: if the estimated worst case gradient norm is large, the training might benefit from clipping in order to reduce the amount of noise added to the gradients, however, when estimated worst case gradient norm is small, the training would benefit of using the estimates and have little added noise.
5. Many experimental details are missing: which architecture/hyperparameters did you use, etc.
6. A very minor comment: when printing on the paper, formulas in green and yellow color are not very visible, try using a different color.

**Limitations:**

yes

---

> ### Author Rebuttal · Authors · 2023-08-08
>
> Thank you for your detailed questions and careful reading.
>
> #### **Remark 1**
>  We agree with your observation that a direct comparison to vanilla DP-SGD is valuable. Please see our general response for preliminary results on Cifar-10 and on two tabular datasets. Note that there isn’t exactly a 1-to-1 mapping between conventional networks and Lipschitz networks, hence some architectures can be advantageous towards one algorithm or the other (see Appendix A1).
>
> #### **Remark 2**
> Note that the competing results we report are not baselines, nor vanilla DP-SGD, but rather the SOTA in the DP training literature. They were obtained using different architectures and relied on ad-hoc pre-processing and algorithmic  techniques on top of vanilla DP-SGD. For instance, in the case of CIFAR10:
> * Abadi et al. rely on a differential private PCA embedding on the images central pixels.
> * Feldman & Zrnic use a Rényi filter to account for individual privacy leakages.
> * Bu et al. leverage the properties of f-Differential Privacy.
> * Chen & Lee use the Armijo condition with noisy gradients and function estimates to find the optimal step size.
> * Papernot et al. uses tempered sigmoids as activation functions.
> * Nasr et al. encode the gradient and appeal to denoising techniques whereas we learn the raw cifar10 from pixels.
> * Yu et al. appeal to the definition of concentrated Differential Privacy for tighter bounds and also implement a dynamic privacy budget allocator to improve model quality.
>
> We note that our framework is theoretically compatible with most of these improvements.
> On MNIST, they are not used as extensively and our approach is on par with the SOTA results.
>
>
> #### **Remark 3**
> In “local strategy” the sensitivity is computed on a per-layer basis, and the noise added is calibrated to each layer individually. In the global strategy, the noise is calibrated to the sensitivity of the whole gradient vector. Please see Appendix A2 p13 (and Algorithm 3 p14) for more details.
>
> #### **Question 1**
>  Each semi-transparent dot on a line is a (val_acc, epsilon) pair measured at the end of an epoch, and over several epochs they correspond to the privacy/accuracy curve of a given set of hyper-parameters. The green line materializes the Pareto front (convex hull of all measurements).
>
> #### **Question 2**
> The papers we cite in Remark 1 propose formal certification methods (i.e bounding boxes, or bounded polytopes, mixed integer nonlinear programming) to bound the output of the network as function of the bounds given in input. Our Algorithm 1 propagates balls, which is significantly faster than formal certification methods, but also less tight. We will clarify this remark.
>
> #### **Question 3**
> Theorem 1 gives an analytical bound on the global sensitivity $\Delta$, as a function of various parameters (maximum bias, smoothness of the loss, maximum input bound, etc). This is useful to get a better understanding of the behavior of the algorithm. Of course in practice we use Algorithm 1 as it is much more practical and versatile since it back-propagates the bounds that have been manually derived in Theorem 1. In particular, the main steps of Algorithm 1 are:
> * Step 2 : the input bound is computed with a forward pass.
> * Step 6 : computes the layer sensitivity from the gradient bounds and from the characteristics of the layer.
> * Step 7 : scalar-scalar backpropagation of the gradient bounds starting from the last layer.
>
> #### **Question 4**
>  We agree that this paragraph could be written more clearly. In a nutshell, we introduce a clipping layer as a final layer in our model that clips the gradients of the loss (wrt the logits) during backpropagation. We gain several advantages :
> * As you said we benefit from a better “gradient to noise” ratio: please see Appendix B5 p18 for a discussion on the bias it induces.
> * As opposed to standard (per-sample) clipping, this process does not slow down the algorithm. Indeed, it is applied only on the last layer with a small dimension (and not on each weight of the full network).
>
> #### **Question 5**
> The architectures used and the range of hyper-parameters are given in Appendix D p25. Please see Figure 10 p26 to see the results broken down per architecture type. Finally, the exact architectures for each task can be found in the folder “experiment” of the Python package. If you believe some important information is missing we will be glad to add it to the supplementary.
>
> #### **Question 6**
> Thank you for the heads up !

---

> > ### Comment · Reviewer_6vik · 2023-08-16
> >
> > I would like to thank the authors for their clarifications that addressed most of my concerns, as well as for extra experiments of DP-SGD. My only remaining question is:
> >
> > Remark 1 & 2. Would it be possible to fix architecture to Lipschitz networks and compare to DP-SGD on these networks? As well as to fix data preprocessing, and compare to the algorithms on the exactly same fixed setting (same data preprocessing & same architecture) ?

---

> > > ### Author Response · Authors · 2023-08-17
> > >
> > > Thank you for your continuing effort in engaging the discussion.
> > >
> > > >  Would it be possible to fix architecture to Lipschitz networks and compare to DP-SGD on these networks?
> > >
> > > Could you clarify your question? From what we understand, you are curious about the performance of DP-SGD on GNP networks? If so, take a look at figure 11 in appendix D3, p26.
> > >
> > > > on the exactly same fixed setting (same data preprocessing & same architecture) ?
> > >
> > > There isn't an exact 1-to-1 mapping between GNP and conventional networks. For example, residuals connections are argued to mitigate the vanishing gradient phenomenon, but this is redundant with the orthogonality condition of GNP networks. Other difficulties exist: the manifold dimension of orthogonal matrices is half the one of all matrices, so at equal width the number of degrees of freedom is actually smaller.
> > >
> > > We attempt a run of Clipless DP-SGD on Cifar-10 with a MLP Mixer, with a standardized Cifar-10 (assuming no privacy cost in computing mean and std, just like in optax tutorial) and we obain:
> > >
> > > | **Epsilon.**| 4.05 | 5.0 | 8.01 | 11.1 | 15.03 |
> > > | ----- | ------ | ------ | ------ | ----- | ---- |
> > > | **Val. Acc.**| 37.6 | 39.3 | 42.0 | 43.2 | 44.5 |

---

> > > > ### Comment · Reviewer_6vik · 2023-08-18
> > > >
> > > > Thank you very much for your reply. I meant comparison the performance of DP-SGD with clipping vs Clippless DP-SGD on the same plot for the same GNP networks.

---

> > > > > ### Author Response · Authors · 2023-08-19
> > > > >
> > > > > > I meant comparison the performance of DP-SGD with clipping vs Clippless DP-SGD on the same plot for the same GNP networks.
> > > > >
> > > > > Indeed, we can merge figure 5 and figure 11. The same architectures have been tested in fig 5 and fig 11. Empirically, the Pareto front is about the same, but the experiments of GNP+Clipping (fig. 11) were much slower to run than GNP+Clipless (fig. 5).

---

> > > > > > ### Comment · Reviewer_6vik · 2023-08-19
> > > > > >
> > > > > > I would like to thank the authors for their replies. I have raised my score by 1 point.

---

### Author Rebuttal · Authors · 2023-08-08

We are happy to see that the reviewers appreciated the originality and potential of our contributions. According to the reviewers, “the concept is **original and potentially significant**” (reviewer mBgX), “the paper is **interesting** and **novel** and **provides direction for new research**” (reviewer 6vik), and “**the idea** of removing clipping as an alternative to clipping itself **is promising**” (reviewer vJyY). They also found that "**the writing is exceptionally lucid**" (reviewer mBgX). We address common concerns below and reply to specific comments in the individual response to each reviewer.

## About experimental results and comparisons to prior work
A concern shared by all reviewers is about the empirical performance of our approach.

We introduce a new baseline approach for DP training, clipless DP-SGD, where the costly per-sample clipping operation used in all current implementations of DP-SGD is replaced by an automatic method that computes upper bounds. Hence, **our core objective is to ensure a fair evaluation** of the idea. We report what can be achieved by this approach in end-to-end training, out of the box, against the SOTA results found in the literature.

We believe there are some pitfalls when it comes to evaluating DP algorithms.
* Some SOTA results used **ad-hoc pre-processing**. For example, the seminal work of Abadi et al (2016) trained the network on the PCA embeddings of Cifar-10. While this yields excellent results, they also set a score that has not been beaten to this day by other “training from scratch” methods (it is known that handcrafted features or pre-training on public data can boost performance in some regimes, see Tramer and Boneh (2020)). Thus hiding the true effectiveness of vanilla DP-SGD on this task.
* Previous papers usually **do not report the Pareto front**: instead, they often report their final accuracy for a few different values of *epsilon* in a table, *after* the extensive fine-tuning of their hyper-parameters. We believe our methodology is more exhaustive and transparent: for example, Figure 4 reports all the runs (including the ones that “failed”) over the broad-range of hyper-parameters (see Appendix D p25 for details).
* Oftentimes, a Lipschitz constrained network and its conventional counterpart may yield completely different results, since they leverage different implicit biases. Hence, it is hard to compare the two algorithms: some architectures may be more beneficial or detrimental to one than the other.

While reaching the SOTA with a new approach is an interesting question per-se, we consider that it is outside the scope of this work. To improve accuracy, we could combine our approach with the many tricks introduced in recent literature (our approach is compatible with many of them, see answer to reviewer **6vik**), and even consider hybrid approaches (as discussed in the paper, and with reviewer **vJyY**). But this would at the same time tend to obfuscate the true nature of our contribution, threaten reproducibility, add complexity and make the evaluation harder. **As we want to foster reproducibility and explorative research, we provide the lip-dp package with reference implementations and documentation.** We keep the framework **as simple as possible**, and as close as possible to the original idea, **so that additional methodological improvements can be evaluated fairly** on top of this baseline. By providing a Python package, we allow every researcher to contribute on the topic and speed-up research advancements.

## Additional experiments

As relevantly noted by reviewer **vJyY**, to complement our original experimental results which compare our approach to the current SOTA in DP training, it is interesting to compare our approach to the fairer baseline of vanilla DP-SGD (with clipping) under a similar computational budget. In the attached PDF, **we have included some preliminary experiments** in this direction:
* A comparison to vanilla DP-SGD with Opacus which uses a ResNet18 with weights pretrained on Imagenet-1k, and by performing a grid search over (learning_rate/maximum gradient norm/batch size) triplets. **We see that our Clipless DP-SGD** (which uses a smaller network trained from scratch) **is close to DP-SGD with pretrained Resnet-18**.
* A comparison to vanilla DP-SGD with Opacus on two tabular datasets (Adult and Breast Cancer). In this case, the architectures are the same except for the activation (see caption of the figure for details). Again, **the results show that our approach is competitive in terms of classification performance.**

In the final version of the paper, we will add the complete Pareto fronts, as done in the original experiments.

## Building Lipschitz networks

The Lipschitz constraints are actually a mild limitation : universal approximation in the set of 1-Lipschitz functions has been proven by the work of Anil et al, and thanks to Bethune et al. we know that 1-Lipschitz NN do not lack expressiveness in classification tasks. The main constraint is that we must rely on projections, and not on re-parametrizations (see the discussion in Appendix A.1.1).

We agree that building Lipschitz constrained networks is not easy (see the discussion in Appendix B.1). Settling the question of the best Lipschitz architecture is still an active research field (see the recent works of Wang and Manchester,  Meunier et al, Araujo et al.), beyond the scope of our work.  We expect our work to evolve together with this field: it will benefit from the progress in Lipschitz networks.

*Wang, R. and Manchester, I., 2023, July. Direct Parameterization of Lipschitz-Bounded Deep Networks. In International Conference on Machine Learning (pp. 36093-36110). PMLR.*

*Araujo, A., Havens, A.J., Delattre, B., Allauzen, A. and Hu, B., 2022, September. A Unified Algebraic Perspective on Lipschitz Neural Networks. In The Eleventh International Conference on Learning Representations.*

---

### Decision · Program_Chairs · 2023-09-21

**Decision:**

Reject

**Comment:**

This paper discusses a new approach to training Differentially Private Deep Neural Networks by analyzing Lipschitz constrained networks to establish sensitivity bounds for each layer, eliminating the need for per-sample gradient clipping.

Although the paper has some merits such as it is very well written, and the approach proposed without using per-sample clipping is interesting, it suffers from a few significant drawback. In particular, one reviewer suggests, and I agree, that this approach is not practical because it suffers from low signal to noise ratio, when the true gradients are potentially tiny. As we know, in practical training the effect of signal to noise ratio will significantly impact the performance of the model obtained, I believe that technique proposed is not practical.